# ReFPO:Flow Matching Policy Gradients is Naturally a Reflow Process

## Abstract

We present Reflow Matching Policy Gradients (ReFPO), a simple and performant RL method that bridges the gap between expressive flow matching policies and real-time control requirements. We uncover a key theoretical property: the gradient updates in Flow Matching Policy Gradients (FPO) act as an implicit advantage-weighted Reflow process a discovery that provides a new geometric foundation for flow-based policy gradients. Building on this insight, ReFPO introduces an explicit geometric regularizer that can be implemented with a single line of code change without incurring additional computational overhead or auxiliary distillation stages. By synergizing advantage-guided updates with path rectification, our method effectively stabilizes the trust-region optimization and enables high-fidelity one-step inference that consistently matches or exceeds multi-step performance. We experimentally demonstrate that ReFPO leads to superior performance and discretization robustness across GridWorld, MuJoCo Playground, and high-dimensional Humanoid Control tasks, providing a scalable and stable foundation for generative policies in complex physical simulations.

## 1. Introduction

The adoption of generative models, particularly flow matching and diffusion, has revolutionized policy representation in reinforcement learning (RL) by enabling the capture of complex, multimodal action distributions. Unlike traditional Gaussian policies, flow-based policies can represent non-convex behaviors essential for high-dimensional tasks. However, their practical utility is severely hampered by their iterative sampling nature. Generating a single action often requires multiple ODE integration steps, introducing

[1]Anonymous Institution, Anonymous City, Anonymous Region, Anonymous Country. Correspondence to: Anonymous Author <anon.email@domain.com>.

Preliminary work. Under review by the International Conference on Machine Learning (ICML). Do not distribute.

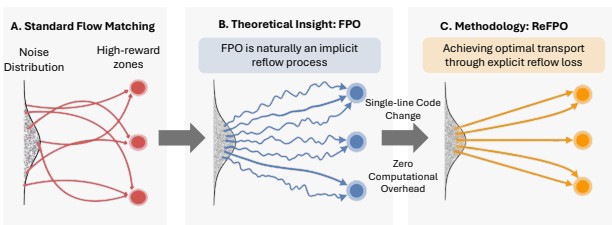

*Figure 1.* **Overview of ReFPO.** (A) Standard Flow Matching often results in curved, inefficient probability paths when mapping noise to high-reward zones. (B) We provide a theoretical insight that FPO is naturally an implicit reflow process, though it lacks explicit path straightening. (C) Our proposed ReFPO achieves optimal transport through an explicit reflow loss, enabling straight sampling paths with a single-line code change and zero additional computational overhead.

a significant latency barrier for real-time robotic deployment (Chi et al., 2025).

To mitigate this, prior acceleration techniques such as Rectified Flow (Liu et al., 2023), Consistency Models (Song et al., 2023), or multi-stage distillation (Yin et al., 2024) have been proposed. While effective in supervised computer vision, directly grafting these methods onto RL often results in cumbersome, multi-stage pipelines that struggle with the non-stationary landscape of online optimization. These "force-fitted" combinations introduce significant computational overhead and additional training complexity, lacking the elegance required for seamless RL integration.

As illustrated in Figure 1, we introduce **Reflow Matching Policy Gradients (ReFPO)**, a framework built on the realization that Flow Matching Policy Gradients (FPO) is, at its core, an advantage-weighted minimization of the Conditional Flow Matching (CFM) loss. We identify that in the context of online RL, where the policy trains on its own self-sampled trajectories, this CFM loss does not perform simple imitation; rather, it realizes a self-iterative Reflow process. This discovery is both natural and elegant: our contribution is the identification that, when embedded in an online RL loop, FPO realizes a nontrivial instance of Reflow dynamics beyond the supervised setting.

However, we uncover a critical stability gap in this implicit process. While the advantage-weighted update straightens flows for high-reward actions, it simultaneously distorts the

flow field for actions with negative advantages. Mathematically, this imbalance induces high curvature in the velocity field and amplifies discretization errors, which destabilizes the trust-region optimization and leads to frequent numerical collapses. To resolve this, ReFPO promotes the CFM loss, a value already computed in the FPO pipeline to an explicit Reflow regularization term. By incorporating this term into the total objective, we enforce a global straightening of the flow field regardless of the advantage sign. This provides a crucial geometric anchor that stabilizes trust-region jitters and yields straighter, more efficient flows.

The primary contributions of this paper are fourfold. First, we provide an implicit Reflow discovery by theoretically establishing that the FPO update rule acts as an implicit advantage-weighted Reflow process within self-sampling regimes, offering a new geometric foundation for flow-based policy gradients. Second, we introduce explicit Reflow Regularization, a stabilizer that ensures global flow consistency and maintains trust-region integrity. Third, we demonstrate robustness to self-collapse, proving that our framework avoids the over-correction and mode-collapse issues typical of standard Reflow by leveraging the dynamic RL advantage signal to preserve expressivity. Finally, we validate our empirical performance across GridWorld, MuJoCo Playground and Humanoid Control benchmarks, where ReFPO achieves perfect results and enables high-fidelity one-step inference without incurring any additional computational cost.

## 2. Related Work

### 2.1. Generative and Flow-based Policies in RL

Early generative policies were primarily developed in the context of offline RL (Janner et al., 2022; Wang et al., 2023; Hansen-Estruch et al., 2023; Park et al., 2025; Zhang et al., 2025a; Wang et al., 2025; Seo et al., 2025) to capture multimodal expert distributions. Recently, the field has increasingly shifted toward online reinforcement learning (Zhang et al., 2025b) to achieve superior performance and environmental adaptability. Initial online methods, such as DDPO (Black et al., 2024), DPPO (Ren et al., 2025), and Flow-GRPO (Liu et al., 2025b), typically treat the iterative denoising process as a multi-step Markov Decision Process (MDP), optimizing each discretized step via standard policy gradients. However, this MDP-based formulation violates the continuous-time nature of the underlying flows and suffers from high computational costs (Li et al., 2025a). To address this, FPO (McAllister et al., 2025) introduced a holistic framework leveraging Conditional Flow Matching (CFM) as a variational lower bound (Kingma & Gao, 2023). While FPO established a tractable gradient framework for flow-based policies, it focuses exclusively on gradient directionality, overlooking the dynamic evolution of the intrinsic flow geometry across RL iterations—a gap that ReFPO aims

to bridge.

### 2.2. Path Rectification and Geometric Stability

The sampling efficiency of continuous-time generative models is fundamentally governed by the straightness of their probability flows (Shankar & Geffner, 2025). While early frameworks established the foundational Reflow (Liu et al., 2023)procedure to minimize transport costs by straightening trajectories, subsequent research has focused on theoretically refining these mappings toward Euclidean geodesics to achieve near-optimal transport (Tong et al., 2024; Kornilov et al., 2024). These advancements have progressively enabled one-step acceleration by unifying consistency-based (Yang et al., 2024; zhangkai wu et al., 2025) objectives with path rectification. However, a persistent challenge in recursive straightening is the risk of model collapse (Zhu et al., 2024a), where the vector field degenerates due to an over-reliance on self-consistency within self-generated data distributions. ReFPO addresses this by framing the policy optimization process as an implicit Reflow paradigm that incorporates a reward-driven corrective force. By synergizing the advantage-guided signal with the flow-matching objective, our method ensures that trajectory rectification is grounded in environmental feedback rather than mere self-imitation. This intrinsic synergy naturally prevents distributional collapse and maintains mode diversity without requiring additional training stages or auxiliary datasets.

### 2.3. High-Fidelity One-Step Inference

Achieving high-fidelity one-step inference has emerged as a pivotal research direction for overcoming the inference latency of generative models, spanning paradigms from supervised fine-tuning to reinforcement learning. In the supervised setting, various approaches have sought to fit one-step trajectories through complex mathematical reformulations to identify optimal transport paths (Frans et al., 2025; Geng et al., 2025; Luo et al., 2025; Zhang et al., 2025c; Liu et al., 2025a; Zhou et al., 2025) , or by compressing pre-trained models via adversarial learning (Cheng et al., 2025; Lin et al., 2025) and knowledge distillation (Zhu et al., 2025; Yin et al., 2024). Similarly, reinforcement learning methods aimed at one-step generation often rely on explicit teacher guidance (Zhu et al., 2024b; Seo et al., 2025), computationally intensive reparameterized gradients (Kornilov et al., 2024; Nguyen & Yoo, 2025; Lv et al., 2025; Chen et al., 2025; Wang et al., 2025) , or additional training stages (Park et al., 2025; Li et al., 2025b). Despite their progress in sampling speed, the common pitfalls of these methods lie in their substantial computational overhead or complex distillation pipelines, which significantly increase the threshold for practical implementation.

In contrast, ReFPO achieves high-fidelity one-step inference through a "straightening-while-training" paradigm. By integrating Reflow regularization into the RL objective, the probability flow is rectified during optimization, enabling $N = 1$ sampling as a natural byproduct without auxiliary distillation or multi-stage pipelines.

## 3. Reflow Matching Policy Gradients

### 3.1. PPO and FPO

The objective of on-policy RL is to learn a policy $\pi_\theta$ that maximizes expected return by increasing the likelihood of actions with positive advantage. Proximal Policy Optimization (PPO) (Schulman et al., 2017) stabilizes policy gradient updates by introducing a clipped likelihood ratio,

$$r(\theta) = \frac{\pi_\theta(a_t \mid o_t)}{\pi_{\theta_{\text{old}}}(a_t \mid o_t)}, \tag{1}$$

and optimizing a clipped surrogate objective that constrains updates to remain within a local trust region. This formulation relies critically on access to tractable action likelihoods. However, the action likelihood is generally intractable to compute in policies based on flow matching.

FPO addresses this limitation by replacing PPO's likelihood ratio with a computable surrogate derived from flow matching objectives. Instead of explicitly evaluating $\pi_\theta(a_t \mid o_t)$, FPO measures how strongly a flow policy transports probability mass toward a sampled action by comparing CFM losses under the current and old policies. Concretely, FPO defines a proxy ratio

$$\hat{r}^{\text{FPO}}(\theta) = \exp\Big( \widehat{\mathcal{L}}^{\text{CFM}}_{\theta_{\text{old}}}(a_t; o_t) - \widehat{\mathcal{L}}^{\text{CFM}}_\theta(a_t; o_t) \Big). \tag{2}$$

and substitutes $\hat{r}^{\text{FPO}}(\theta)$ directly into the PPO clipped surrogate objective.

This substitution is theoretically motivated by the connection between flow matching losses and the evidence lower bound (ELBO) of the induced policy distribution. For flow and diffusion models, minimizing the denoising or flow matching loss is equivalent to maximizing the ELBO up to a constant and thus increasing the likelihood of the target action. The exponential difference of flow matching losses therefore acts as a proxy for the likelihood ratio used in PPO.

With this replacement, FPO preserves the core structure of PPO, including clipping, advantage weighting, and on-policy updates, while enabling the training of expressive flow-based policies without explicit likelihood computation.

### 3.2. FPO as an Implicit Online Reflow Process

We show that FPO implements an implicit advantage-weighted Reflow update in an online, self-sampled setting,

rather than merely inheriting the straightening property of CFM in supervised learning.

A key distinction from standard flow matching formulations is that, in FPO, the action pairs $(\epsilon, a_1)$ used to define the CFM objective are not drawn from a fixed dataset. Instead, they are generated on-policy by the flow policy itself. As a result, the learning dynamics form a closed loop: the policy both induces the training distribution and is updated based on it. This self-sampling property places FPO squarely in the regime studied by Reflow, where rectification is applied recursively to model-generated samples.

To make this connection explicit, consider the unclipped FPO surrogate objective

$$\mathcal{L}(\theta) = -\mathbb{E}_t\big[\hat{r}_t(\theta) A_t\big], \tag{3}$$

where the proxy ratio is defined as

$$\hat{r}_t(\theta) = \exp\Big(\mathcal{L}^{\text{CFM},\theta_{\text{old}}}(a_t; o_t) - \mathcal{L}^{\text{CFM},\theta}(a_t; o_t)\Big). \tag{4}$$

Since the old-policy loss $\mathcal{L}^{\text{CFM},\theta_{\text{old}}}(a_t; o_t)$ is treated as a stop-gradient, taking the gradient with respect to $\theta$ then yields

$$\frac{\partial}{\partial \theta}\big(\hat{r}_t(\theta) A_t\big) = -A_t \hat{r}_t(\theta) \, \nabla_\theta \mathcal{L}^{\text{CFM},\theta}(a_t; o_t). \tag{5}$$

Therefore, for a single sample, positive advantages drive updates along $-\nabla_\theta \mathcal{L}^{\text{CFM}}$, whereas negative advantages invert the gradient direction.

Furthermore, we unpack the CFM loss per action sample:

$$\mathcal{L}^{\text{CFM},\theta}(a_t; o_t) = \mathbb{E}_{\tau,\epsilon}\big[\| v_\theta(a_{\tau,t}, \tau; o_t) - (a_t - \epsilon) \|^2\big], \tag{6}$$

with noisy intermediate $a_{\tau,t} = \alpha_\tau a_t + \sigma_\tau \epsilon$. Minimizing the per-sample CFM loss forces the model velocity field $v_\theta(\cdot, \tau)$ at every intermediate $a_{\tau,t}$ to align with the same vector $a_t - \epsilon$. Therefore, minimizing this loss aligns the velocity field along straight segments connecting self-generated noise–action pairs.

Taken together, these observations show that FPO performs an advantage-weighted, online Reflow process, in which the policy repeatedly rectifies its own induced trajectories under task-driven feedback. We therefore refer to this perspective as ReFPO (Reflow Matching Policy Gradients ), highlighting that FPO can be viewed as an implicit, advantage-weighted rectified flow procedure.

### 3.3. Enhancing Stability via Explicit Reflow Regularization

PPO's clipping enforces a local trust region by requiring the likelihood-ratio change to be small. If clipping is frequently triggered, it ceases to be a protective mechanism

---

**Algorithm 1** Reflow Matching Policy Gradients (ReFPO)

---

**Require:** Policy parameters $\theta$, value parameters $\phi$, clip parameter $\epsilon$, MC samples $N_{\text{mc}}$, Reflow coefficient $\lambda$

**while** not converged **do**

    Collect trajectories using current flow policy and compute advantages $\hat{A}_t$

    For each action, store $N_{\text{mc}}$ timestep-noise pairs $\{(\tau_i, \epsilon_i)\}$ and compute $\ell_\theta(\tau_i, \epsilon_i)$

    $\theta_{\text{old}} \leftarrow \theta$

    **for** each optimization epoch **do**

        Sample mini-batch from collected trajectories

        **for** each state-action pair $(o_t, a_t)$ and corresponding MC samples $\{(\tau_i, \epsilon_i)\}$ **do**

            Compute flow matching loss $\ell_\theta(\tau_i, \epsilon_i)$ using stored $(\tau_i, \epsilon_i)$

            $\hat{r}_\theta \leftarrow \exp\left(-\frac{1}{N_{\text{mc}}}\sum_{i=1}^{N_{\text{mc}}}(\ell_\theta(\tau_i, \epsilon_i) - \ell_{\theta_{\text{old}}}(\tau_i, \epsilon_i))\right)$

            $\mathcal{L}^{\text{Reflow}}(\theta) \leftarrow \frac{1}{N_{\text{mc}}}\sum_{i=1}^{N_{\text{mc}}}\ell_\theta(\tau_i, \epsilon_i)$ {Reuse $\ell_\theta$ from above}

            $L^{\text{FPO}}(\theta) \leftarrow \min(\hat{r}_\theta \hat{A}_t, \text{clip}(\hat{r}_\theta, 1 \pm \epsilon)\hat{A}_t)$

            $L^{\text{ReFPO}}(\theta) \leftarrow L^{\text{FPO}}(\theta) + \lambda \mathcal{L}^{\text{Reflow}}(\theta)$

        **end for**

        $\theta \leftarrow \text{Optimizer}(\theta, \nabla_\theta \sum L^{\text{ReFPO}}(\theta))$

    **end for**

    Update value function parameters $\phi$ like standard PPO

**end while**

---

and becomes a persistent hard truncation, reducing the number of effective samples. In FPO, the clipping condition can be written in terms of the change in the conditional flow-matching loss,

$$\left|\Delta\mathcal{L}^{\text{CFM}}\right| := \left|\mathcal{L}^{\text{CFM},\theta} - \mathcal{L}^{\text{CFM},\theta_{\text{old}}}\right| \le \log(1+\varepsilon). \quad (7)$$

To characterize how parameter updates affect this constraint, we examine the first-order sensitivity of $\Delta\mathcal{L}^{\text{CFM}}$ with respect to $\delta\theta$. Specifically, we perform a first-order Taylor expansion of $\mathcal{L}^{\text{CFM},\theta}$ around $\theta_{\text{old}}$, which yields

$$\Delta\mathcal{L}^{\text{CFM}} \approx \mathbb{E}_\tau\big[2\langle r_\tau, \partial_\theta v \, \delta\theta + \partial_a v \, \delta a_\tau\rangle\big], \quad (8)$$

where

$$r_\tau = v_{\theta_{\text{old}}}(a_\tau, \tau) - (a_\tau - \epsilon_\tau), \quad (9)$$

and $\delta a_\tau$ denotes the perturbation of the flow trajectory induced by the parameter change.

The trajectory perturbation $\delta a_\tau$ follows the variational equation associated with the flow ODE,

$$\frac{\text{d}}{\text{d}\tau}\delta a_\tau = J_v(a_\tau, \tau)\,\delta a_\tau + \partial_\theta v \, \delta\theta, \quad (10)$$

where $J_v$ is the Jacobian of the velocity field with respect to the state.

If the flow exhibits nontrivial curvature, quantified by a lower bound on the Jacobian norm $\|J_v\| \gtrsim \kappa > 0$,

Grönwall-type bounds give $\|\delta a_\tau\| = O(e^{\kappa\tau})\|\delta\theta\|$ and hence

$$\left|\Delta\mathcal{L}^{\text{CFM}}\right| = O\left(\frac{e^\kappa - 1}{\kappa}\,\|\delta\theta\|\right) \sim O(e^\kappa\|\delta\theta\|). \quad (11)$$

Therefore, even when parameter updates $\|\delta\theta\|$ are small, high curvature in the learned flow can exponentially amplify changes in the CFM loss, leading to frequent violations of the clipping constraint.

To mitigate this failure mode, we propose a Reflow regularizer (unweighted CFM loss) to encourage straight, path-optimal velocities explicitly:

$$\mathcal{L}_{\text{ReFlow}}(\theta) = \mathbb{E}_{\tau,\epsilon}\big[\|v_\theta(a_\tau, \tau) - (a_1 - \epsilon)\|_2^2\big]. \quad (12)$$

$$\mathcal{L}(\theta) = \mathbb{E}_t\big[\mathcal{L}^{\text{FPO}}(\hat{r}_t(\theta), A_t)\big] + \lambda\mathcal{L}_{\text{ReFlow}}(\theta), \quad (13)$$

Driving $\mathcal{L}_{\text{ReFlow}}$ down biases the learned vector field toward the rectified-flow solution $v^\star(a_\tau, \tau) = a_1 - \epsilon$, for which $\partial_a v^\star = 0$ (i.e. $J_v \approx 0$). In that regime $\|\delta a_\tau\| = O(\tau\|\delta\theta\|)$ and one recovers the benign bound $\left|\Delta\mathcal{L}^{\text{CFM}}\right| = O(\|\delta\theta\|)$.

Although FPO can be interpreted as an advantage-weighted implicit Reflow procedure, the explicit Reflow regularization introduced here serves a distinct and complementary role. The instability analyzed above does not stem from insufficient straightening of high-advantage trajectories, but from the geometric sensitivity of the new-old policy difference, where small parameter updates can induce large changes in the CFM loss due to flow curvature.

Crucially, the Reflow regularizer is applied after the policy ratio is formed and does not reweight trajectories by advantage. It therefore does not compete with the reward-driven objective, but instead acts as a purely geometric constraint that suppresses curvature in the learned velocity field. While the advantage-weighted FPO term determines which trajectories are reinforced, the Reflow regularization controls how sensitively the loss responds to parameter updates.

As a result, straightening low-advantage trajectories does not dilute the reward signal: these trajectories are not promoted by the policy objective, but merely stabilize the training process by preventing curvature-induced amplification.

### 3.4. Synergizing Advantage Guidance and Self-Consistency

The training of Reflow improves sampling efficiency by repeatedly aligning the learned vector field with straight-line transport. However, when applied in a self-sampling regime, this rectification becomes increasingly self-consistent: the model is trained on trajectories generated by its own policy. In this setting, minimizing the flow matching loss encourages the vector field to align with its induced transport directions, rather than correcting toward unexplored regions

of the target distribution. As a result, Reflow tends to reinforce existing modes while suppressing diversity, ultimately leading to degenerate solutions.

ReFPO mitigates this failure mode through joint optimization with the advantage-guided policy objective. As defined in Eq. (13), the FPO term introduces an external, reward-driven signal that is not self-generated by the current policy. This signal selectively amplifies high-advantage trajectories, counteracting the over-regularized self-alignment of Reflow and maintaining effective exploration. Consequently, ReFPO preserves the efficiency benefits of rectified flows while avoiding collapse induced by excessive self-consistency.

## 4. Experiments

We evaluate ReFPO across three distinct domains: a multi-goal Grid World to visualize distribution straightness (Brockman et al., 2016; Towers et al., 2024), MuJoCo Playground (Todorov et al., 2012; Zakka et al., 2025) for standard continuous control benchmarks, and high-dimensional Humanoid Control to test tracking under-conditioned control signals.

### 4.1. GridWorld

We evaluate ReFPO in a $25 \times 25$ GridWorld with bifurcated sparse rewards to investigate its capacity for modeling complex action distributions and its robustness to discretization. As a baseline, we compare our results with vanilla FPO (McAllister et al., 2025), which demonstrated the ability of flow-based policies to capture multimodal behaviors.

**Multimodality and Path Optimality.** In Figure 2, we visualize the denoising trajectories for both FPO and ReFPO using $N = 10$ Euler steps. Our results confirm that ReFPO faithfully preserves the multimodal nature of the policy, successfully bifurcating the probability flow at the central saddle point toward the top and bottom goals. Crucially, we observe a distinct path optimality phenomenon in ReFPO. While FPO trajectories exhibit stochastic curvature—a byproduct of the unconstrained coupling in early RL iterations—ReFPO "rewires" these flows into straighter, more direct paths. This behavior aligns with the theoretical properties of the reflow process, which drives the velocity field toward Euclidean geodesics, effectively minimizing the transport cost during policy refinement.

**High-Fidelity One-Step Inference.** A core objective of ReFPO is to enable low-latency inference without compromising performance. Figure 2 compares the one-step ($N = 1$) denoising results. Notably, we observe that vanilla FPO is already capable of completing the task with a single Euler step, which provides strong empirical evidence for

our central hypothesis: the iterative policy gradient updates in FPO naturally act as a self-rectification process.

However, this "inherent" reflow effect in FPO is often unrefined, leading to sub-optimal coupling. As evidenced by the action vector plots in Figure 2, while FPO reaches the goal, its single-step actions exhibit stochastic fluctuations and noticeable directional drift. This instability arises because FPO does not explicitly minimize the path curvature, making its one-step approximation sensitive to the discretization error of the underlying curved flow.

In contrast, ReFPO leverages the explicit Reflow loss to achieve a more rigorous geometric alignment. The resulting action samples exhibit significantly higher spatial consistency and lower variance. By "straightening" the probability flow, ReFPO ensures that even a zero-order integration ($N = 1$) aligns closely with the high-fidelity ODE trajectory, demonstrating superior robustness and reliability for real-time deployment.

### 4.2. MuJoCo Playground

We evaluate ReFPO on a suite of 10 continuous control tasks adapted from the DeepMind Control Suite (Tassa et al., 2018; Tunyasuvunakool et al., 2020) within the MuJoCo Playground environment. These tasks, ranging from CheetahRun to ReacherHard, serve as a benchmark for high-dimensional control and reward density. Following our theoretical analysis in Section 3.3, these experiments focus on the trade-off between sampling efficiency and policy performance.

**Experimental Setup and Metrics.** Following the protocol established in FPO, we utilize the Adam (Kingma, 2014) optimizer with a batch size of 1024 and 16 updates per batch. To observe the asymptotic behavior and long-term stability of the learned flow, we extend the training budget to 100M environment steps. Policies are evaluated using both $N = 1$ and $N = 10$ inference steps to measure discretization robustness.

Beyond cumulative reward, we introduce two key metrics for geometric analysis: (i) Straightness Error ($\downarrow$), which calculates the Mean Squared Error (MSE) between the learned velocity field $v_\theta$ and the ideal constant velocity ($a_1 - a_0$) over 100 integration steps. A smaller error indicates a more linear path. (ii) Explosion Rate ($\downarrow$), defined as the frequency of samples where the log-likelihood difference $|\mathcal{L}_{new} - \mathcal{L}_{old}|$ exceeds a threshold of 3. This threshold corresponds to a policy ratio $\hat{r}(\theta) = \exp(\mathcal{L}_{new} - \mathcal{L}_{old})$ exceeding $\exp(3) \approx 20.1$, a value that signifies a severe departure from the local trust region. This metric effectively captures the numerical instability during the exponential calculation of the proxy ratio. We find that a regularization coefficient of $\lambda = 0.04$ yields the optimal balance.

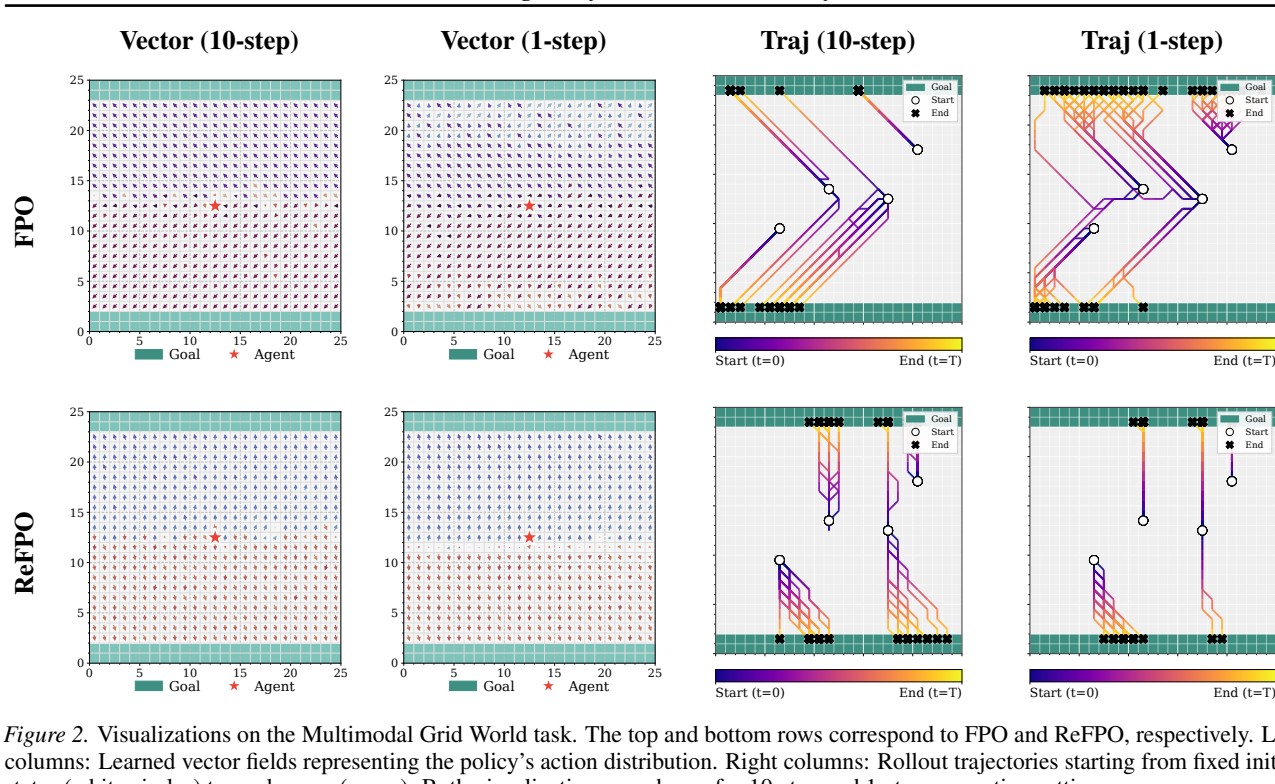

*Figure 2.* Visualizations on the Multimodal Grid World task. The top and bottom rows correspond to FPO and ReFPO, respectively. Left columns: Learned vector fields representing the policy's action distribution. Right columns: Rollout trajectories starting from fixed initial states (white circles) to goal zones (green). Both visualizations are shown for 10-step and 1-step generation settings.

*Table 1.* **ReFPO variant comparison.** Cumulative rewards (mean $\pm$ std), straightness, and stability across different configurations. ReFPO* denotes the variant with $\lambda = 0.04$.

| METHOD | REWARD | REWARD(1STEP) | STRAIGHTNESS ERROR | EXPLOSION RATE |
|---|---|---|---|---|
| PPO | 622$\pm$192 | / | / | / |
| FPO | 641$\pm$140 | 565$\pm$160 | 0.0475 | 0.00365 |
| REFPO* | **686$\pm$139** | **690$\pm$139** | 0.0116 | **0.00189** |
| REFPO,$\lambda = 0.02$ | 644$\pm$132 | 556$\pm$160 | **0.00472** | 0.00377 |
| REFPO,$\lambda = 0.08$ | 646$\pm$141 | 652$\pm$151 | 0.0161 | 0.00569 |
| REFPO,$\lambda = 0.12$ | 609$\pm$147 | 601$\pm$155 | 0.0193 | 0.00330 |

**Results and Dynamics Analysis.** The training trajectories across ten tasks (Figure 3) and the aggregate metrics (Table 1) collectively demonstrate that ReFPO fundamentally optimizes the policy's underlying geometry. A salient macro-level observation is the remarkable consistency between one-step ($N = 1$) and multi-step ($N = 10$) performance in ReFPO. While vanilla FPO manifests a substantial discretization gap—evidenced by the prominent shaded areas between its trajectories in Figure 3—ReFPO's curves exhibit near-identical progression. This robustness is a direct consequence of the minimized Straightness Error, where the alignment of the policy flow toward Euclidean geodesics eliminates the cumulative integration drift. Consequently, ReFPO preserves high-fidelity control capabilities without the iterative sampling overhead typically required by generative policies.

Furthermore, the training dynamics reveal that ReFPO

achieves superior optimization stability compared to the high-volatility curves of FPO. In complex environments like BallInCup and FingerSpin, FPO experiences erratic fluctuations and occasional performance collapses, whereas ReFPO maintains a smoother, monotonic improvement. This enhanced robustness stems from the suppression of the Explosion Rate. By straightening the flow, ReFPO constrains the Jacobian spectral norm of the velocity field, preventing the policy ratio $\hat{r}(\theta)$ from frequently breaching the trust-region boundaries ($|\Delta\mathcal{L}| > 3$). This reduction in trust-region violations ensures that the optimization process is shielded from gradient spikes, allowing for higher asymptotic rewards and ensuring the numerical integrity of the policy updates throughout the training loop.

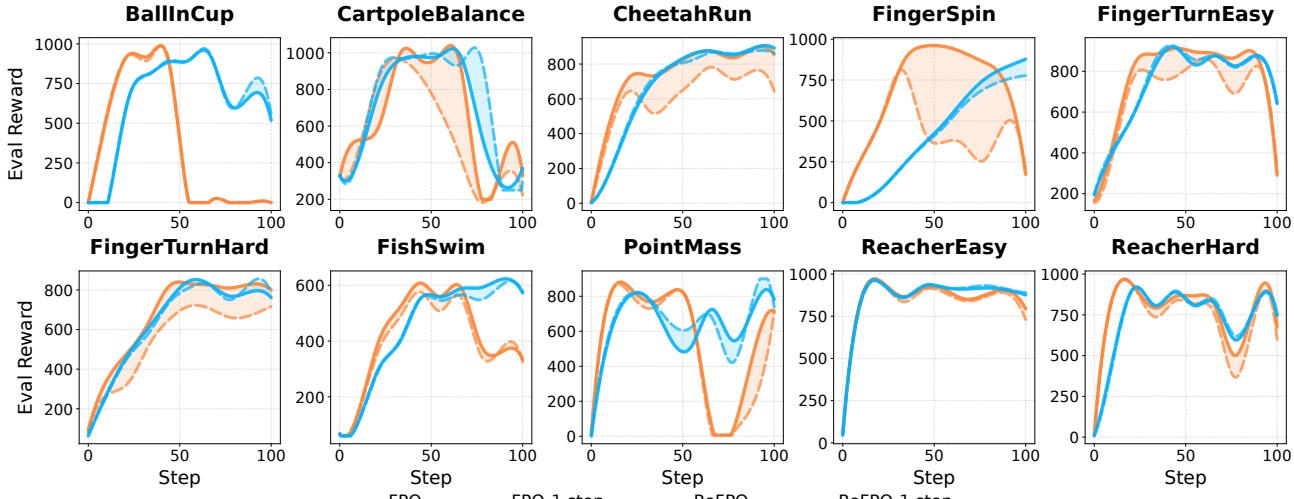

*Figure 3.* Comparison of FPO and ReFPO on DM Control Suite. The curves illustrate the evaluation rewards over 100M environment steps. The shaded regions represent the performance gap between the 10-step and 1-step generation for each method. A narrower shaded area indicates higher consistency between the accelerated 1-step policy and the standard 10-step baseline.

## 4.3. Humanoid Control

Physics-aware humanoid control represents a significantly higher-dimensional challenge than standard MuJoCo benchmarks, serving as a stringent test for the scalability of action flows. We evaluate ReFPO on a motion-capture (MoCap) tracking task where a simulated SMPL-based agent must reproduce diverse reference motions from the AMASS dataset (Mahmood et al., 2019). This environment requires the policy to maintain physical balance while performing precise, high-degree-of-freedom coordination, making it a critical arena for assessing discretization robustness and geometric stability. As shown in Figure 4a, ReFPO successfully tracks the reference motion while FPO fails to maintain balance. More examples are available in the supplementary materials.

**Implementation Details and Metrics.** Our simulated agent, featuring 24 actuated joints (72 DoF) organized in a kinematic tree, is trained within the Isaac Gym (Makoviychuk et al., 2021) physics engine. We utilize the Puffer-PHC implementation (Luo et al., 2023) as our baseline, where the policy receives both proprioceptive state information and goal-conditioned targets. To investigate the interaction between information density and flow geometry, we explore three conditioning regimes: (i) Full-joint conditioning, (ii) Root+Hands conditioning, and (iii) Root-only conditioning. The policy is optimized for 100M environment steps using the imitation reward structure established in (Peng et al., 2018). Performance is quantified using Success Rate (↑), Alive Duration (↑), and the global MPJPE (↓) to measure tracking precision.

**Results and Qualitative Analysis.** The experimental results in Table 2 reveal several key insights into the behavior of action flows in high-dimensional humanoid control. As illustrated by the inference time comparison in Figure 4, the 1-step variants ($N = 1$) offer a substantial reduction in latency compared to the 10-step versions, a speedup that is of paramount importance for responsive control in real-time physical simulations. This empirical efficiency directly supports a core argument in our methodology: since both FPO and ReFPO are fundamentally grounded in the Reflow formulation, they inherently possess the capability for high-fidelity single-pass inference. Especially in these stable, long-horizon training environments, the 1-step scores are consistently on par with, or even superior to, the 10-step results. This suggests that for a well-rectified flow, additional ODE integration steps may introduce unnecessary discretization noise, whereas a single pass provides a cleaner and more coherent control signal.

However, we also observe that the performance margin between ReFPO and the vanilla FPO baseline is relatively modest in these specific tasks. This can be attributed to the nature of the humanoid environment used here, which features dense rewards and stable dynamics. Under such conditions, the advantage signal naturally guides the policy toward a Reflow-like convergence even without explicit geometric constraints, leading to relatively straight trajectories with few "geometric breakthroughs" during training. Nevertheless, ReFPO consistently maintains a stable lead over FPO.

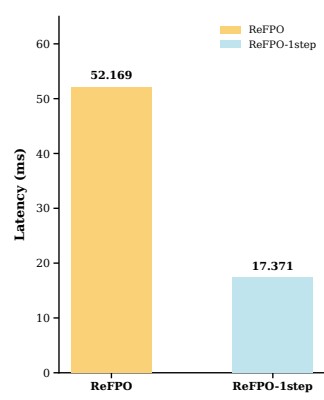

*(a)* Performance on Humanoid Control.

*(b)* Inference latency comparison.

*Figure 4.* Performance and efficiency analysis. (a) Comparison of FPO and ReFPO on Humanoid Control; (b) Action generation latency, where ReFPO-1step achieves significantly lower inference time.

*Table 2.* **Quantitative evaluation on Humanoid Control tasks.** We report performance across three levels of conditioning difficulty using Success Rate ($\uparrow$), Alive Duration ($\uparrow$), and Mean Per Joint Position Error (MPJPE, $\downarrow$). The 1-step variants ($N = 1$) explicitly demonstrate the discretization robustness of ReFPO* compared to the vanilla FPO baseline.

| METHODS | GOAL CONDITIONING | SUCCESS RATE ($\uparrow$) | ALIVE DURATION ($\uparrow$) | MPJPE ($\downarrow$) |
|---|---|---|---|---|
| GAUSSIAN PPO | ALL JOINTS | **98.7**% | **200.46** | **31.62** |
| FPO | ALL JOINTS | 96.2% | 197.64 | 42.65 |
| FPO-1STEP | ALL JOINTS | 96.9% | 198.25 | 41.20 |
| REFPO* | ALL JOINTS | 96.6% | 198.03 | 41.00 |
| REFPO*-1STEP | ALL JOINTS | 97.3% | 198.47 | 39.49 |
| GAUSSIAN PPO | ROOT + HANDS | 46.5% | 142.50 | 97.65 |
| FPO | ROOT + HANDS | 71.0% | 171.68 | 63.88 |
| FPO-1STEP | ROOT + HANDS | 70.8% | 171.37 | 64.96 |
| REFPO* | ROOT + HANDS | 72.7% | **174.98** | 62.68 |
| REFPO*-1STEP | ROOT + HANDS | **74.4**% | 174.83 | **60.60** |
| GAUSSIAN PPO | ROOT | 29.8% | 114.06 | 123.70 |
| FPO | ROOT | 55.0% | 147.94 | 73.55 |
| FPO-1STEP | ROOT | 55.3% | 147.90 | 72.65 |
| REFPO* | ROOT | 58.2% | 156.37 | 70.97 |
| REFPO*-1STEP | ROOT | **58.7**% | **156.45** | **70.89** |

This indicates that while FPO benefits from some inherent straightening, it remains susceptible to trust-region instabilities that can hinder optimal convergence. ReFPO, by virtue of its more rigorous geometric regularization, achieves a "straighter" and more robust flow, ultimately converging to a superior fixed point compared to the baseline.

## 5. Conclusion

We introduced ReFPO, a framework that synergizes advantage-guided policy updates with explicit reflow process. By uncovering the intrinsic "straightening-while-training" property of the FPO gradient and augmenting it with a single-line geometric regularizer, we demonstrate that generative policies can be linearized during the main RL loop with negligible overhead. Specifically, the straightened probability flow allows the agent to produce high-fidelity

actions in a single Euler step, bypassing the high computational cost of iterative ODE solvers. Consequently, the model retains the ability to capture complex, multi-modal distributions while achieving the execution speed of simple Gaussian policies. This approach effectively bridges the gap between the high expressivity of flow-matching models and the low-latency requirements of real-time control.

As a closing remark, we would like to reiterate the primary contribution of ReFPO—its simplicity. In a field where acceleration often necessitates complex multi-stage distillation or auxiliary extra calculation, ReFPO achieves high-fidelity one-step inference through a minimal modification of the core objective. Given that online RL is notoriously sensitive to implementation details, providing a robust, single-stage method for geometric stability is a significant step toward making generative policies a practical standard in robotics.

## Impact Statement

This paper presents work whose goal is to advance the field of Machine Learning. There are many potential societal consequences of our work, none which we feel must be specifically highlighted here.

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

## A. Code and Supplementary Videos

The source code for ReFPO is included in the supplementary materials to ensure the reproducibility of all experimental results.

Additionally, we provide supplementary videos demonstrating the performance of ReFPO and FPO across MuJoCo Playground and Humanoid Control tasks. These videos showcase the high-fidelity trajectories achieved during one-step inference ($N = 1$) and provide direct comparisons with the baseline methods.

## B. ReFPO Derivation

We analyze how a small parameter perturbation $\delta\theta := \theta - \theta_{\text{old}}$ can induce a large change in the conditional flow-matching (CFM) loss

$$\Delta\mathcal{L}^{\text{CFM}} := \mathcal{L}^{\text{CFM},\theta} - \mathcal{L}^{\text{CFM},\theta_{\text{old}}},$$

thereby violating the PPO-style clipping condition even when $\|\delta\theta\|$ is small. The argument proceeds in three steps: (i) a first-order Taylor expansion of $\mathcal{L}^{\text{CFM}}$ at $\theta_{\text{old}}$; (ii) characterization of the induced perturbation of the flow trajectory; (iii) exponential amplification bounds under high curvature of the learned velocity field.

**CFM loss and first-order expansion.** Recall the per-sample CFM objective

$$\mathcal{L}^{\text{CFM},\theta}(a_t; o_t) = \mathbb{E}_{\tau,\epsilon}\left[\left\|v_\theta(a_{\tau,t}, \tau; o_t) - (a_t - \epsilon)\right\|^2\right], \tag{14}$$

where the noisy intermediate state is explicitly constructed as $a_{\tau,t} = \alpha_\tau a_t + \sigma_\tau \epsilon$. For notational brevity, we suppress the dependence on $(a_t, o_t)$ whenever clear.

We view $\mathcal{L}^{\text{CFM},\theta}$ as a composition

$$\theta \mapsto v_\theta(\cdot, \tau) \mapsto \mathcal{L}^{\text{CFM}}(v_\theta),$$

and consider the variation of the loss induced by a small perturbation $\delta\theta$. Assuming that $v_\theta$ is continuously differentiable in both $\theta$ and its state argument, the first-order Taylor expansion of $\mathcal{L}^{\text{CFM},\theta}$ about $\theta_{\text{old}}$ yields

$$\Delta\mathcal{L}^{\text{CFM}} = \left\langle D_\theta \mathcal{L}^{\text{CFM}}(\theta_{\text{old}}), \delta\theta\right\rangle + o(\|\delta\theta\|), \tag{15}$$

where $D_\theta$ denotes the Fréchet derivative. Applying the chain rule to the squared-error structure of the loss gives the explicit linearization

$$\Delta\mathcal{L}^{\text{CFM}} \approx \mathbb{E}_\tau\left[2\left\langle r_\tau, \ \partial_\theta v\, \delta\theta + \partial_a v\, \delta a_\tau\right\rangle\right], \tag{16}$$

where

$$r_\tau := v_{\theta_{\text{old}}}(a_\tau, \tau) - (a_\tau - \epsilon), \tag{17}$$

and $\partial_\theta v$ and $\partial_a v$ denote the Jacobians of $v_\theta(a, \tau)$ with respect to $\theta$ and $a$, evaluated at $(\theta_{\text{old}}, a_\tau)$.

The two terms in the linearization have distinct interpretations: the first corresponds to the *direct* sensitivity of the velocity field to parameter perturbations, while the second captures the *indirect* effect through perturbations of the state along the flow, represented by $\delta a_\tau$.

**Perturbation of the flow trajectory.** Although the CFM loss in the algorithm is evaluated on explicitly constructed interpolants $a_{\tau,t}$, to analyze sensitivity we interpret the learned velocity field $v_{\theta_{\text{old}}}$ as inducing a continuous flow via the ODE

$$\frac{\mathrm{d}}{\mathrm{d}\tau}a_\tau = v_{\theta_{\text{old}}}(a_\tau, \tau).$$

Under a parameter perturbation $\delta\theta$, the induced deviation of this flow satisfies the variational equation obtained by differentiating the ODE with respect to $\theta$:

$$\frac{\mathrm{d}}{\mathrm{d}\tau}\delta a_\tau = J_v(a_\tau, \tau)\,\delta a_\tau + \partial_\theta v(a_\tau, \tau)\,\delta\theta, \tag{18}$$

where $J_v(a_\tau, \tau) := \partial_a v_{\theta_{\text{old}}}(a_\tau, \tau)$ is the Jacobian of the velocity field along the nominal trajectory. The second term acts as a forcing that continuously injects parameter-dependent deviations into the flow.

**Exponential amplification under high curvature.** Assume that $J_v$ and $\partial_\theta v$ are bounded and continuous. Let $\Phi(\tau, s)$ denote the state-transition operator associated with $J_v$. By the variation-of-constants formula,

$$\delta a_\tau = \int_0^\tau \Phi(\tau, s)\, \partial_\theta v(a_s, s)\, \delta\theta\, \mathrm{d}s.$$

Let $L := \sup_{s \in [0,1]} \|J_v(a_s, s)\|$ and $B := \sup_{s \in [0,1]} \|\partial_\theta v(a_s, s)\|$. Applying Grönwall's inequality yields

$$\|\delta a_\tau\| \le B\, \|\delta\theta\| \int_0^\tau e^{L(\tau - s)}\, \mathrm{d}s = B\, \|\delta\theta\| \frac{e^{L\tau} - 1}{L}. \tag{19}$$

Hence $\|\delta a_\tau\| = O(e^{L\tau}\|\delta\theta\|)$, revealing exponential amplification when the flow Jacobian norm $L$ is large.

Substituting (19) into the linearized loss variation and applying Cauchy–Schwarz gives

$$\left|\Delta\mathcal{L}^{\mathrm{CFM}}\right| \lesssim 2\,\mathbb{E}_\tau\Big[\|r_\tau\|\big(\|\partial_\theta v\|\|\delta\theta\| + \|\partial_a v\|\|\delta a_\tau\|\big)\Big]$$
$$= O\Big(\frac{e^L - 1}{L}\, \|\delta\theta\|\Big), \tag{20}$$

where all multiplicative constants depend only on bounded moments of $\|r_\tau\|$, $\|\partial_\theta v\|$, and $\|\partial_a v\|$.

**Implication for clipping.** When the velocity field exhibits large curvature (large $L$), the factor $(e^L - 1)/L$ becomes exponentially large. As a consequence, even small parameter updates $\delta\theta$ can induce large changes in $\mathcal{L}^{\mathrm{CFM}}$, violating the clipping condition

$$\left|\Delta\mathcal{L}^{\mathrm{CFM}}\right| \le \log(1 + \varepsilon),$$

and causing frequent activation of the PPO-style clip. This explains why, in high-curvature regimes, FPO-style updates can become unstable despite conservative parameter steps.

## C. Training Settings

### C.1. GridWorld

All GridWorld experiments were conducted using an **NVIDIA GeForce RTX 4090 GPU**. To ensure a rigorous and fair comparison, we maintained identical hyperparameters for both the vanilla FPO baseline and our proposed ReFPO method, with the only exception being the Reflow regularization coefficient $\lambda$ utilized in ReFPO. A comprehensive summary of these hyperparameters is provided in Table C.3.

*Table C.3.* Hyperparameters for GridWorld Experiments

| HYPERPARAMETER | VALUE |
|---|---|
| OPTIMIZER | ADAM |
| LEARNING RATE | $3 \times 10^{-4}$ |
| TOTAL TIMESTEPS | 260,000 |
| TIMESTEPS PER BATCH | 2048 |
| MINIBATCH SIZE | 341 (APPROX. 2048/6) |
| UPDATES PER ITERATION | 10 |
| MAX EPISODE LENGTH | 200 |
| DISCOUNT FACTOR ($\gamma$) | 0.99 |
| GAE PARAMETER ($\lambda_{GAE}$) | 0.98 |
| PPO CLIP RANGE ($\epsilon$) | 0.2 |
| MAX GRAD NORM | 0.5 |
| MC SAMPLES FOR CFM ($N_{mc}$) | 50 |
| REFLOW COEFFICIENT | 0.1 (0.0 FOR FPO) |
| ODE SOLVER | EULER |
| INFERENCE STEPS ($N$) | 1, 10, 20 |

## C.2. MuJoCo Playground

For the MuJoCo Playground benchmarks, we evaluated our methods across 10 continuous control tasks using an **NVIDIA GeForce RTX 4090 GPU**. All models were trained for a total of 100 million environment steps to observe asymptotic performance and flow stability. For PPO, we utilized a standard Gaussian policy as a baseline, training it for the same 100 million environment steps to align the experimental configuration across all methods. Similar to the GridWorld experiments, the hyperparameters for FPO and ReFPO are identical except for the Reflow regularization coefficient $\lambda$ and Output mode. Regarding the Output mode, it is worth noting that while the theoretical formulations of both FPO and ReFPO are inherently based on velocity prediction, the vanilla FPO baseline adopts a modified output mode specifically tuned to maximize cumulative rewards in certain environments. However, our empirical analysis indicated that such a modification is suboptimal for the ReFPO framework and compromises the stability of the learned flows. Therefore, we strictly adhere to the standard velocity prediction in our method to maintain consistency with the underlying flow-matching objective, ensuring a more stable and theoretically grounded training process.

For ReFPO, we applied a constant regularization coefficient of $\lambda = 0.04$, which was empirically found to provide the best balance between path straightness and policy expressivity in these high-dimensional tasks. The detailed hyperparameter configurations are summarized in Table C.4.

*Table C.4.* Hyperparameters for PPO and FPO/ReFPO.

(a) PPO

| Hyperparameter | Value |
| --- | --- |
| Discount factor ($\gamma$) | 0.995 |
| GAE $\lambda$ | 0.95 |
| Value loss coefficient | 0.25 |
| Entropy coefficient | 0.01 |
| Reward scaling | 10.0 |
| Normalize advantage | True |
| Normalize observations | True |
| Action repeat | 1 |
| Unroll length | 30 |
| Batch size | 1024 |
| Number of minibatches | 32 |
| Number of updates per batch | 16 |
| Number of timesteps | 100M |
| Policy network | MLP (4, 32) |
| Value network | MLP (5, 256) |
| Optimizer | Adam |

(b) FPO / ReFPO

| Hyperparameter | Value |
| --- | --- |
| Discount factor ($\gamma$) | 0.995 |
| GAE $\lambda$ | 0.95 |
| Value loss coefficient | 0.25 |
| Clipping $\epsilon$ | 0.05 |
| Reward scaling | 10.0 |
| Reflow Coefficient | 0.04 (ReFPO) |
| Normalize advantage | True |
| Normalize observations | True |
| Flow steps ($T$) | 10 |
| Samples per action | 8 |
| Output mode | velocity/eps |
| Time embed dim | 8 |
| Unroll length | 30 |
| Batch size | 1024 |
| Number of minibatches | 32 |
| Number of updates per batch | 16 |
| Number of timesteps | 100M |
| Policy network | MLP (4, 32) |
| Value network | MLP (5, 256) |
| Optimizer | Adam |

## C.3. Humanoid Control

All humanoid control experiments were conducted using an **NVIDIA A100 GPU**, with each individual training run requiring approximately **14 hours** to complete. To ensure a rigorous and fair comparison, we maintained identical hyperparameters for both the vanilla FPO baseline and our proposed ReFPO method, with the only exception being the Reflow regularization coefficient $\lambda$ (associated with the Reflow loss) utilized exclusively in ReFPO. A comprehensive summary of these hyperparameters is provided in Table C.5.

# D. Detailed Results on MuJoCo Playground

In this section, we provide the detailed evaluation results across ten MuJoCo Playground environments. The performance metrics for FPO and ReFPO are comprehensive summarized in Table D.6 and Table D.7, respectively.

*Table C.5.* Hyperparameters for humanoid control.

| Hyperparameter | Value | Hyperparameter | Value |
|---|---|---|---|
| | *Policy Settings* | | |
| Hidden size | 512 | Solver step size | 0.1 |
| Action perturbation std | 0.05 | Target KL divergence | None |
| Number of environments | 4096 | Normalize advantage | True |
| Reflow Coefficient | 0.1 (0.0 for FPO) | | |
| | *Training Settings* | | |
| Batch size | 131072 | Minibatch size | 32768 |
| Learning rate | 0.0001 | LR annealing | False |
| LR decay rate | 1.5e-4 | LR decay floor | 0.2 |
| Update epochs | 4 | L2 regularization coef. | 0.0 |
| GAE lambda | 0.2 | Discount factor ($\gamma$) | 0.98 |
| Clipping coefficient | 0.01 | Value function coefficient | 1.2 |
| Clip value loss | True | Value loss clip coefficient | 0.2 |
| Max gradient norm | 10.0 | Entropy coefficient | 0.0 |
| Discriminator coefficient | 5.0 | Bound coefficient | 10.0 |

*Table D.6.* Detailed Results on MuJoCo Playground (FPO)

| Environment | Reward mean | Reward std | Reward mean_1step | Reward std_1step | Straightness mse | Clipping Epsilon (MSE) |
|---|---|---|---|---|---|---|
| BallInCup | 368.94 | 45.93 | 370.07 | 45.38 | 0.0037 | 0.00068501 |
| CartPoleBalance | 358.95 | 72.12 | 314.59 | 66.14 | 0.0749 | 0.00471780 |
| CheetahRun | 783.27 | 75.57 | 643.56 | 76.94 | 0.0374 | 0.00071751 |
| FingerSpin | 699.81 | 53.44 | 415.83 | 69.14 | 0.1946 | 0.00433034 |
| FingerTurnEasy | 770.78 | 213.13 | 692.09 | 316.59 | 0.0066 | 0.00016701 |
| FingerTurnHard | 710.62 | 304.10 | 582.97 | 383.59 | 0.0021 | 2.63e-05 |
| FishSwim | 512.33 | 177.24 | 481.91 | 181.27 | 0.0532 | 2.77e-08 |
| PointMass | 528.11 | 84.22 | 459.37 | 68.29 | 0.0914 | 0.02583412 |
| ReacherEasy | 883.78 | 162.03 | 861.99 | 169.60 | 0.0049 | 7.09e-08 |
| ReacherHard | 793.57 | 211.45 | 739.56 | 221.52 | 0.0059 | 2.07e-05 |
| **AVERAGE** | **641.02** | **139.92** | **556.19** | **159.85** | **0.0475** | **0.00364989** |

*Table D.7.* Detailed Results on MuJoCo Playground (ReFPO)

| Environment | Reward mean | Reward std | Reward mean_1step | Reward std_1step | Straightness mse | Clipping Epsilon (MSE) |
|---|---|---|---|---|---|---|
| BallInCup | 667.66 | 149.91 | 680.05 | 143.76 | 0.0069 | 0.0089154 |
| CartPoleBalance | 709.15 | 45.95 | 756.54 | 39.96 | 0.0027 | 0.00079641 |
| CheetahRun | 726.20 | 38.44 | 722.16 | 56.80 | 0.0222 | 0.000064723 |
| FingerSpin | 472.23 | 14.62 | 439.63 | 19.62 | 0.0406 | 0.00032414 |
| FingerTurnEasy | 757.07 | 260.81 | 763.74 | 260.60 | 0.0070 | 0.0036175 |
| FingerTurnHard | 703.97 | 261.09 | 704.68 | 260.34 | 0.0039 | 0.0020739 |
| FishSwim | 479.39 | 159.93 | 468.73 | 169.15 | 0.0241 | 0.000072129 |
| PointMass | 668.38 | 129.54 | 676.72 | 123.02 | 0.0024 | 0.002375845 |
| ReacherEasy | 904.44 | 102.75 | 910.91 | 96.47 | 0.0019 | 0.000016144 |
| ReacherHard | 768.11 | 222.43 | 773.56 | 221.23 | 0.0039 | 0.00067742 |
| **AVERAGE** | **685.66** | **138.55** | **689.67** | **139.09** | **0.0116** | **0.001893361** |

