# OpenReview forum: "ReFPO:Flow Matching Policy Gradients is Naturally a Reflow process"
_ICML.cc/2026/Conference — Submitted to ICML 2026_

### Official Review · Reviewer_KzDd · 2026-03-13

**Soundness:** 3
**Presentation:** 3
**Significance:** 3
**Originality:** 3
**Overall Recommendation:** 4
**Confidence:** 5

**Summary:**

This paper addresses a practical limitation of flow-based policies in RL: their iterative ODE-based inference is too slow for real-time deployment, and existing acceleration methods (distillation, consistency training) are cumbersome to integrate into online RL loops. The authors make a key theoretical observation, that Flow Matching Policy Gradients (FPO) is, at its core, an implicit advantage-weighted Reflow process, and build on this to propose ReFPO (Reflow Matching Policy Gradients), which introduces an explicit geometric Reflow regularizer as a single-line code change requiring zero additional computational overhead.

The authors show that FPO's per-sample CFM loss trains the velocity field to align with straight segments connecting self-generated noise-action pairs, which is precisely a self-iterative Reflow dynamic. Second, they identify a stability gap: advantage-weighting distorts the velocity field for negative-advantage actions, inducing high curvature that exponentially amplifies CFM loss changes under small parameter updates (proven via Grönwall's inequality), causing frequent PPO clipping violations. Third, the Reflow regularizer, an unweighted CFM loss term, enforces global flow straightness, driving the flow jacobian toward zero, which reduces the amplification factor from exponential to linear. Empirically, ReFPO is validated on a 25×25 GridWorld, 10 MuJoCo Playground tasks, and a 72-DoF Humanoid Control task, consistently outperforming FPO in cumulative reward, discretization robustness, and training stability, while achieving near-identical N=1 and N=10 inference performance enabling high-fidelity one-step inference without distillation.

**Compliance With Llm Reviewing Policy:**

Affirmed.

**Final Justification:**

The rebuttal provided by the authors have clarified my questions, so I will keep my positive score of 4.

**Key Questions For Authors:**

1. In the joint objective (Eq. 13), $L_{ReFlow}$ is applied uniformly regardless of advantage sign. For a trajectory sampled by the current policy with negative advantage $A_t < 0$, the FPO term ($L^{FPO}$) drives the velocity *away* from that action, while $L_{ReFlow}$ simultaneously drives the velocity *toward* the straight path to that action. Can the authors provide a formal argument or at minimum an ablation comparing trajectory-advantage-weighted Reflow (weighting $L_{ReFlow}$ by $|A_t|$) showing that this interference is negligible? How does the relative magnitude of $\lambda$ to typical $|A_t|$ values determine the degree of interference?

2. The exponential amplification bound (Eq. 19, 20) requires $||J_v|| ≥ κ > 0$. Can the authors report the spectral norm of $J_v$ (or its Frobenius norm as a proxy) during training for both FPO and ReFPO across representative tasks? This would verify that (a) the high-curvature regime is actually realized during FPO training and (b) the ReFPO regularizer successfully reduces $J_v$ toward zero. Without this, the bound is a conditional statement about a regime whose practical occurrence is unconfirmed.

3. The related work cites Flow-GRPO (Liu et al., 2025b) and Reinflow (Zhang et al., 2025b) as directly relevant concurrent approaches that also target online flow-matching policy optimization. Neither is included as a baseline. Can the authors provide at least one empirical comparison on a shared benchmark task? Specifically: (a) does ReFPO achieve better 1-step vs. multi-step parity than Reinflow, which explicitly targets straight trajectory learning via fine-tuning? (b) Does ReFPO's single-stage approach outperform Flow-GRPO's MDP-based treatment without flow geometry?

4. Appendix C.2 states that vanilla FPO uses a modified output mode specifically tuned to maximize cumulative rewards in certain environments, while ReFPO strictly uses standard velocity prediction. How many of the 10 MuJoCo tasks use this modified output mode for FPO? Does removing this modification from FPO (i.e., evaluating both FPO and ReFPO with identical standard velocity prediction) change the relative performance gap? If so, how much of ReFPO's advantage is attributable to the regularizer versus this output mode difference?

**Limitations:**

Yes

**Strengths And Weaknesses:**

**Strengths:**
- The implicit Reflow identification for FPO (Section 3.2) is a non-trivial and elegant theoretical result. The key steps are clearly executed: (i) unpacking the per-sample CFM loss (Eq. 6) reveals that minimizing it trains the velocity to point from noisy interpolants toward self-generated actions; (ii) since these action pairs are generated on-policy (not from a fixed dataset), the policy induces its own training distribution, placing FPO squarely in the self-iterative Reflow regime described by Liu et al. (2023).
- ReFPO derivation in Appendix B is rigorous and very well written: a first-order Taylor expansion of $\Delta L^{CFM}$ (Eq. 15–16) → variational ODE for the flow trajectory perturbation (Eq. 18) → Grönwall's inequality yields $||\delta a_\tau|| = O(e^(L_\tau)||\delta \theta||)$ (Eq. 19) → the full loss perturbation bound $|\Delta L^{CFM}|$ = $O(\frac{e^L - 1}{L} · ||\delta \theta||)$ (Eq. 20). This cleanly explains why high-curvature flows cause PPO clipping violations even for small parameter steps, providing a principled causal account of FPO's instability.
- Figure 2 (GridWorld vector field and trajectory visualizations) provides compelling visual evidence: ReFPO's straighter trajectories and tighter 1-step/10-step agreement are immediately apparent.

**Weaknesses:**
- The paper claims that the Reflow regularizer $L_{ReFlow}$ "does not reweight trajectories by advantage" and acts as a "purely geometric constraint that does not compete with the reward-driven objective." However, in the joint objective (Eq. 13), both terms are minimized simultaneously in a single gradient step. For a trajectory with *negative* advantage (which $L^{FPO}$ is trying to suppress), $L_{ReFlow}$ still pushes the velocity to align with the straight path from noise to that negative-advantage action. This could partially counteract the advantage-based suppression signal. The paper does not prove that this interference is negligible, nor does it provide an ablation that tests whether trajectory-weighted Reflow would be more or less stable.

- The Grönwall-based amplification bound (Eq. 19) is activated only when the Jacobian of the velocity field has a meaningful spectral lower bound $κ$. This is assumed, not derived or verified. In practice, $J_v$ depends on the learned network architecture and training regime. The paper never reports measured Jacobian norms during training, so it is unclear whether the high-curvature regime is actually realized or how severe it is relative to the theoretical worst case.

- Appendix C.2 acknowledges that the vanilla FPO baseline adopts a modified output mode specifically tuned to maximize cumulative rewards in certain environments, while ReFPO strictly uses standard velocity prediction. This means FPO receives task-specific output mode tuning that ReFPO deliberately forgoes for theoretical consistency. Since this modification affects reward performance in some environments, results where FPO underperforms ReFPO may partially reflect this difference rather than pure flow geometry, and this is not discussed in the main text.

- The concurrent Reinflow (Zhang et al., 2025b) also fine-tunes flow matching policies with online RL targeting flow straightness. The differentiation from this work deserves explicit discussion, particularly regarding what ReFPO achieves that Reinflow does not.

---

> ### Author Rebuttal · Authors · 2026-03-31
>
> We sincerely thank the reviewer for the positive assessment of our theoretical analysis and visual evidence, as well as for the constructive questions on the empirical scope of the work. We have added supplementary results on our anonymous website ([supplementary link](https://anonymous.4open.science/w/icml_rebuttal-1E27/)), including Extra Table 1 and Extra Table 2, which we reference below where relevant.
>
>
>
> ## Question 1
>
> We thank the reviewer for the question. ReFlow is intentionally sign-agnostic because curvature harms optimization stability regardless of advantage sign. Our goal is therefore to make the learned velocity field straighter for both positive- and negative-advantage trajectories. This is important because the FPO surrogate is clipped based on the change in the CFM proxy, and excessive flow curvature makes the policy update overly sensitive, causing the surrogate to be clipped more frequently.
>
>
> The role of the ReFlow regularizer is therefore auxiliary: it acts as a geometric stabilizer rather than a dominant optimization signal. In our formulation, the total update is:
> $$g_{\mathrm{total}} = g_{\mathrm{FPO}} + \lambda g_{\mathrm{ReFlow}}$$
> and we operate in the regime where the gradient ratio $\rho$ is small:
> $$\rho = \frac{\lambda \|g_{\mathrm{ReFlow}}\|}{\|g_{\mathrm{FPO}}\|} \ll 1.$$
> Hence, the ReFlow term does not materially alter which actions are favored by the policy objective; instead, it provides a small but important regularizing force that suppresses curvature and improves optimization stability.
>
> We conducted an additional experiment on trajectory-advantage-weighted ReFPO (weighting $\mathcal{L}_{ReFlow}$ by $|A_t|$) to clarify our design choice and address the reviewer's curiosity. As shown in Extra Table 1, while this variant (with $\lambda = 0.04$) shows some improvement over vanilla FPO, it remains inferior to standard ReFPO in overall score. Most importantly, it fails to achieve the "stable straight-flow" property of ReFPO, where one-step reward matches ten-step reward. Trajectory-advantage weighting entangles geometric regularization with reward signals, breaking the uniformity required for global flow stability.
>
>
> ## Question 2
>
> Yes. We now report the Jacobian explosion rate, ratio explosion rate, mean Jacobian norm, and 95th percentile of the Jacobian norm in Extra Table 2. For brevity, please see Reviewer 2, Question 3.
>
>
> ## Question 3
>
> We initially did not include Flow-GRPO[1] and Reinflow[2] as baselines because they are primarily designed as post-training RL methods , whereas FPO and ReFPO are "train-from-scratch" approaches. Nevertheless, we have now deployed both on the MuJoCo Playground for a rigorous comparison. We ensured all methods: (1) use identical model parameters and environment steps; (2) follow the algorithmic formulations and setups detailed in their original papers; and (3) tuned hyperparameters (learning rate, clip ratio) to maximize performance.
>
> For the main comparison, all methods are evaluated on the same 0-100 epoch training window. We additionally report the 100-200 epoch results for Flow-GRPO and Reinflow to test whether longer training changes the conclusion; it does not. While we also attempted to accelerate their train-from-scratch performance using larger learning rates, this consistently led to collapse, so we used the largest learning rates that still avoided divergence. Furthermore, while Reinflow demonstrates good one-step parity , Flow-GRPO's one-step reward dropped to zero. We conclude that Reinflow and Flow-GRPO are better suited for large-scale post-training, while ReFPO is optimized for training smaller-scale models from scratch for tasks like humanoid control.
>
>
> ## Question 4
>
> We emphasize that ReFPO strictly utilizes standard velocity prediction ($u$) across all 10 MuJoCo tasks, as well as in GridWorld and Humanoid Control. In contrast, while vanilla FPO uses standard velocity for GridWorld and Humanoid control , it adopts a modified output mode in MuJoCo Playground to maximize cumulative rewards. This modification ($u$ supervised by $\epsilon$) essentially defines a time-weighted loss:
> $$\mathcal{L}_{\mathrm{modified}} = \|x_1 - \hat{x}_1\|^2 = (1-t)^2 \|u - \hat{u}\|^2$$
> As $t \to 1$, this weighting substantially downweights the loss, which may empirically improve stability, but it also departs from the standard CFM parameterization and changes the training objective in a task-specific way. In this sense, the observed stability gain of vanilla FPO may partially come from output-mode tuning rather than from a principled control of flow geometry itself.
>
> We therefore view ReFPO as the more theoretically standard approach, since it preserves the original flow-matching objective and attributes the improvement to an explicit geometric mechanism.
>
> [1] Flow-GRPO: Training Flow Matching Models via Online RL, NeurIPS'2025
>
> [2] ReinFlow: Fine-tuning Flow Matching Policy with Online Reinforcement Learning, NeurIPS'2025

---

> > ### Author Rebuttal · Reviewer_KzDd · 2026-04-02
> >
> > Thank you for your detailed clarification. My concerns have been addressed and I will keep my positive score.

---

### Official Review · Reviewer_YkYn · 2026-03-13

**Soundness:** 3
**Presentation:** 2
**Significance:** 3
**Originality:** 3
**Overall Recommendation:** 4
**Confidence:** 3

**Summary:**

The submission identifies a problem in a previous method, FPO, that the implicit recitification can destabilize the training. A regularization term that explicitly encourages rectification is proposed to mitigate this problem. Empirical evidence shows that this method improves over the previous FPO.

**Compliance With Llm Reviewing Policy:**

Affirmed.

**Final Justification:**

The added experiments in the rebuttal phase significantly strengthen the soundness of the empirical evaluation of the work. The authors' clarification on the broad applicability of the curvature-driven failure mode should be of broad interest in the flow policy community. Therefore, I recommend this paper to be accepted.

**Key Questions For Authors:**

1. In equation 6, should the expectation be taken only with respect only to $\tau$ instead of both $\tau$ and $\epsilon$, because $\epsilon$ is tied with the corresponding $a_t$ that generates it? I am not entirely sure if I understood correctly, but if so, the presentation in this subsection can be improved by explicitly pointing out $\epsilon$ and $a_t$ are paired and then the objective forms a rectified flow-like loss (up to a reweighting factor).

2. I don't fully understand this claim:
> Crucially, the Reflow regularizer is applied after the policy
... ratio is formed and does not reweight trajectories by advantage. It therefore does not compete with the reward-driven
objective, but instead acts as a purely geometric constraint
that suppresses curvature in the learned velocity field.

(line 195-200)

According to equation 5, the FPO objective is equivalent to a reweighted CFM loss (where the reweighting factor has stopped gradient), which is also proportional to the ReFlow regularizer. Can you explain the above claim more specifically?

3. Do you have empirical evidence that the curvature-induced clipping is exactly what destabilizes FPO training? Currently, the submission seems to only present some theoretical motivation, so empirical studies on the detailed training dynamics and failure modes of FPO would more strongly support findings. (See weakness 1)

4. The rectified-flow style straightening effect is widely present in advantage-reweighted flow matching style policies, such as [1]. Is the curvature-driven clipping effect present in other reweighting-style flow matching RL methods? If so, does the current theoretical insight carry on to those cases?  If the analysis and the proposed solution can be extended to a wider family of algorithms, the contribution of the submission would be larger.

[1] Online Reward-Weighted Fine-Tuning of Flow Matching with Wasserstein Regularization, ICLR'25

**Limitations:**

Yes

**Strengths And Weaknesses:**

Strengths:
1. The work is well-motivated with some theoretical insight (curvature in the flow VF frequently triggers clipping). The proposed solution is simple and yet seems effective, as backed by experiments across different benchmarks.

Weaknesses:
1. Empirical evidence is less convincing, since baselines in the experiments are insufficient. Only PPO, FPO, and the proposed ReFlow are evaluated.
2. The claimed failure mode of FPO is not thoroughly demonstrated in experiments. This weakens the motivation of the entire proposed algorithm that straightness contributes to the desired training dynamics, because it is not directly supported that it is this effect that causes the failure modes of FPO.
3. The phenomenon of curvature-driven clipping is itself interesting. However, I am not sure if this is widely interested to the community, since the analysis and empirical evaluation in the submission are limited to the FPO algorithm (which is not a peer-reviewed publication).


Minor:

l18: ...as an implicit advantage weighted Reflow process [add a comma here?] a discovery that ...

---

> ### Author Rebuttal · Authors · 2026-03-31
>
> We thank the reviewer for recognizing the motivation and practical promise of our method, as well as for the thoughtful questions that helped us clarify the presentation. We have also added supplementary experimental results on our anonymous website ([supplementary link](https://anonymous.4open.science/w/icml_rebuttal-1E27/)), including Extra Table 1 and Extra Table 2, which we will refer to below where relevant.
>
>
> ## Question 1
>
> Thank you for pointing out this important detail regarding the coupling of variables. We agree that in our reinforcement learning setting, the action $a_t$ is not independently sampled but is generated from the noise variable $\epsilon$ through the flow policy. Consequently, the intermediate state $a_\tau$ is also deterministically constructed from the same $\epsilon$.
>
> To clarify, the per-action sample CFM loss in Equation 6 should be interpreted as an expectation over the interpolation time $\tau$:
>
> $$
> \mathcal{L}^{CFM,\theta}(a_t; o_t) = E_\tau [ \| v_\theta(a_{\tau,t}, \tau; o_t) - (a_t - \epsilon) \|^2 ].
> $$
>
>
> We will revise the manuscript to make this dependency explicit and eliminate any ambiguity regarding the construction of $a_{\tau,t}$ as $\alpha_\tau a_t + \sigma_\tau \epsilon$.
>
>
>
> ## Question 2
>
> We wish to clarify that the ReFlow regularizer is independent of the reward-based reweighting, rather than the base CFM loss itself. While the FPO term reweights samples by the clipped proxy ratio $\hat{r}_t$ and the advantage $A_t$, the ReFlow regularizer applies the CFM objective uniformly across all trajectories without any advantage factor.
>
> Thus, it does not alter which trajectories are reinforced by the environment's reward signal. Instead, it serves as an auxiliary geometric prior that suppresses curvature in the velocity field globally, ensuring that even low-advantage trajectories remain straight to stabilize the training dynamics.
>
>
> ## Question 3
>
> To address this question, we report four statistics in Extra Table 2: the Jacobian explosion rate, the ratio explosion rate, the mean Jacobian norm, and the $95^{\text{th}}$ percentile of the Jacobian norm.
>
> Among these, the most important quantities are the explosion rates, because they directly measure how often training enters the numerically unstable regime. We define a Jacobian explosion event as $\|J\| > 3$, and a ratio explosion event as $|\Delta \mathcal{L}_{\mathrm{CFM}}| > 3$. We choose this threshold because these terms enter the sensitivity bound exponentially, so a value of 3 already corresponds to a very large amplification scale (roughly $e^3 \approx 20$), which is precisely the regime where optimization becomes unstable in practice.
>
> We additionally report the mean and the $95^{\text{th}}$ percentile of the Jacobian norm as complementary statistics: the mean reflects the overall scale, while the $95^{\text{th}}$ percentile captures the high-curvature tail without being overly affected by a few extreme outliers. Across all these metrics, ReFPO is consistently lower than FPO, with the clearest gap appearing in the explosion rates.
>
> This is also consistent with the reward-collapse behavior shown in Figure 3, where vanilla FPO suffers significant degradation in tasks such as `FingerSpin`, `FishSwim`, and `FingerTurnHard` during later training. Finally, regarding Weakness 1, we have supplemented the experimental results for Flow-GRPO[1], Reinflow[2], and Advantage-Weighted ReFPO in Extra Table 1; detailed discussion is provided in our response to Reviewer KzDd's Question 3.
>
>
> ## Question 4
>
> We thank the reviewer for relating our work to other advantage-reweighted flow-matching methods. Our point is not that path straightening is unique to ReFPO, but that the curvature-driven sensitivity bottleneck we analyze is a more general property of self-sampled flow policies.
>
> As derived in Section 3.3 and Appendix B, policy sensitivity is governed by the variational equation of the flow:
>
> $$\frac{d}{d\tau} \delta a_\tau = J_v(a_\tau, \tau) \delta a_\tau + \partial_\theta v(a_\tau, \tau) \delta \theta$$
>
> This confirms that trajectory perturbations $\delta a_\tau$ are exponentially amplified by the Jacobian of the velocity field $J_v$. In any online RL method learning from self-sampled data, high curvature (a large Jacobian norm) leads to numerical instability, e.g., [3].
>
> In PPO-style objectives like FPO, this manifests as frequent clipping violations. In other methods, it results in high gradient variance or sensitivity to learning rates. ReFPO acts as a "stability layer" that complements global distribution alignment by suppressing local geometric sensitivity, making it a robust addition to any online flow-matching framework.
>
> [1] Flow-GRPO: Training Flow Matching Models via Online RL, NeurIPS'2025
>
> [2] ReinFlow: Fine-tuning Flow Matching Policy with Online Reinforcement Learning, NeurIPS'2025
>
> [3] Online Reward-Weighted Fine-Tuning of Flow Matching with Wasserstein Regularization, ICLR'25

---

> > ### Author Rebuttal · Reviewer_YkYn · 2026-04-02
> >
> > I acknowledge the authors for the reply. My concerns in questions are resolved but those mentioned in weaknesses remain. Can the authors comment on or clarify those points?

---

> > > ### Author Response · Authors · 2026-04-02
> > >
> > > We sincerely thank the reviewer for the follow-up and for acknowledging that our responses to the questions were helpful.
> > >
> > > ## Weakness 1
> > >
> > > We agree that the original experimental section was limited in its baseline coverage. During the rebuttal period, we therefore added two recent and directly relevant baselines, ReinFlow[1] and Flow-GRPO[2], and report their results in Extra Table 1 below.
> > >
> > > The key takeaway from these added comparisons is that, in this train-from-scratch setting, ReFPO remains clearly stronger than both methods, especially in 1-step generation. ReinFlow shows partial 1-step parity but lower overall returns, while Flow-GRPO performs substantially worse and even collapses to near-zero 1-step reward in our setting.
> > >
> > > ### Extra Table 1
> > >
> > > | Method | Reward 10steps (0-100epochs) | Reward 1steps (0-100epochs) | Reward 10steps (100-200epochs) | Reward 1steps (100-200epochs) |
> > > |---|---:|---:|---:|---:|
> > > | ppo | 622±192 | \ | \ | \ |
> > > | fpo | 641±140 | 565±160 | \ | \ |
> > > | refpo | 686±139 | 690±139 | \ | \ |
> > > | reinflow | 416±195 | 339±194 | 523±188 | 357±178 |
> > > | flow-grpo | 475±194 | 0.000144±0.000218 | 507±235 | 0.000150±0.000226 |
> > > | Trajectory-advantage-weighted ReFPO | 671.12±160.8 | 639.52±165.74 | \ | \ |
> > >
> > > ## Weakness 2
> > >
> > > We agree that this point deserves direct empirical support. During the rebuttal period, we added Extra Figure 1 on our supplementary page ([supplementary link](https://anonymous.4open.science/w/icml_rebuttal-1E27/)) https://anonymous.4open.science/w/icml_rebuttal-1E27/ to visualize this failure mode directly. The visualization videos provided in our Supplementary Material also support the same observation.
> > >
> > > As shown in Extra Figure 1, in `BallInCup`, `FingerSpin`, and `PointMass`, the reward collapse of FPO is accompanied by a simultaneous explosion in both the exponential Jacobian norm and the policy ratio. In contrast, ReFPO keeps both quantities much smaller and the reward trajectory correspondingly more stable. We believe this provides direct empirical evidence that the instability of FPO is closely associated with curvature-induced clipping and ratio explosion.
> > >
> > > Extra Table 2 provides the corresponding aggregate statistics, where ReFPO is consistently lower than FPO in Jacobian explosion rate, ratio explosion rate, mean Jacobian norm, and 95th percentile of the Jacobian norm. This is fully consistent with our response to Question 3.
> > >
> > > ### Extra Table 2
> > >
> > > | Method | Jacobian explosion rate | Ratio explosion rate | Mean Jacobian norm | 95th percentile of Jacobian norm |
> > > |---|---:|---:|---:|---:|
> > > | refpo | 0.005654 | 0.00189 | 0.4881 | 0.9067 |
> > > | fpo | 0.008724 | 0.00365 | 0.5551 | 1.219 |
> > >
> > > ## Weakness 3
> > >
> > > We agree that this is an important question. As clarified in our response to Question 4, our claim is not that the straightening effect itself is unique to FPO, but that the curvature-driven sensitivity mechanism identified here is likely to extend more broadly to self-sampled, advantage-reweighted flow-matching policies. In particular, this is closely related to the broader class of online reward-weighted flow-matching methods discussed there[3], rather than being specific only to FPO.
> > >
> > > We therefore view FPO not as the only setting where this idea matters, but as the cleanest case study in which this mechanism can be isolated, analyzed, and directly connected to optimization instability. From this perspective, the contribution of the paper is not only an improvement to FPO, but also the identification of a more general geometric instability mechanism together with a corresponding stabilization strategy. We agree that extending the same analysis and regularization idea to a wider class of online flow-matching RL methods is an important next step.
> > >
> > > Finally, while the review notes that FPO[4] was not peer-reviewed at the time of submission, it has since been accepted to ICLR 2026. We therefore believe the contribution is of broader interest not only because it improves FPO itself, but because it identifies a potentially general instability mechanism and a corresponding stabilization strategy for online flow-matching RL.
> > >
> > > We sincerely thank the reviewer again for the constructive follow-up. We hope that, with the question-related concerns resolved and the weakness-related concerns now further clarified through the added experiments and failure-mode visualization, the overall contribution of the paper can be reassessed more favorably.
> > >
> > > [1] Flow-GRPO: Training Flow Matching Models via Online RL, NeurIPS'2025
> > >
> > > [2] ReinFlow: Fine-tuning Flow Matching Policy with Online Reinforcement Learning, NeurIPS'2025
> > >
> > > [3] Online Reward-Weighted Fine-Tuning of Flow Matching with Wasserstein Regularization, ICLR'25
> > >
> > > [4] Flow Matching Policy Gradients, ICLR'26

---

### Official Review · Reviewer_EFJy · 2026-03-14

**Soundness:** 1
**Presentation:** 1
**Significance:** 1
**Originality:** 1
**Overall Recommendation:** 2
**Confidence:** 5

**Summary:**

This paper points out that the recently proposed Flow Matching Policy Gradients (FPO) can be viewed as a reflow procedure and propose a modification to FPO that they claim improves stability enabling more reliable few-step action generation. To validate these claims they evaluate their method on various continuous control tasks.

**Compliance With Llm Reviewing Policy:**

Affirmed.

**Final Justification:**

Given the acknowledgement that the theoretical motivation of their method is wrong / misleading I maintain my score. While I appreciate the honest acknowledgement, this is not a minor fix --- the sensitivity analysis was the central justification for the proposed regularizer and occupied a significant portion of the paper. The post-hoc rationalizations offered in the second rebuttal, while more carefully stated, are new arguments and amount to a substantially different paper. Revising the core motivation for a method is outside the scope of what can be addressed in a single round of reviews.

For what it's worth, I believe the actual explanation is much simpler than either the original or revised derivations: when FPO encounters negative advantage, it maximizes the CFM loss, pushing $v_\theta$​ away from the straight-line path connecting $\epsilon \to a_t$. Also not that this is completely unconstrained. The added CFM regularizer simply prevents this divergence from becoming too large. This is a one-sentence motivation that requires no sensitivity analysis or Jacobian arguments.

**Key Questions For Authors:**

Some questions were provided in the above section although the answer to these questions probably won't change my evaluation of the paper.

**Limitations:**

There's no discussion about the limitations of the proposed method.

**Strengths And Weaknesses:**

Although it wasn't pointed out in the original FPO paper that coupling the noise can be viewed as performing reflow, anyone familiar with flow matching / diffusion would have identified this immediately. I don't think this insight in and of itself is novel but rather what you do with this insight that should be the primary contribution.

The only problem is that the proposed modification to FPO has nothing to do with reflow and the premise for this modification is false. The authors claim the learned policy flow with FPO will have high curvature. In the derivation, this hinges upon the term $\partial_a v \delta a_\tau$ and analyzing the sensitivity of $a_\tau$ with respect to the policy parameters $\theta$. The only problem is that this term is exactly $0$ because the action $a_\tau$ doesn't depend on $\theta$ as it's generated through the linear interpolation of $\epsilon$ and $a_1$. Although $a_1$ depends on the parameters we aren't taking the gradient through the sampling procedure. Therefore, the implication from the sensitivity analysis of the ODE is false. In fact, you could make this same flawed argument for the regular CFM loss, it's not even specific to FPO.

On top of this, the empirical methodology needs improvement. Some of the concerns I have:
- How many seeds were these experiments run over? The only type of statistical guarantees for the empirical results are in Table 1 which shows massively overlapping confidence intervals.
- When presenting the humanoid results you say: "To investigate the interaction between information density and flow geometry," as the motivation for trying different conditioning schemes but it's not even clear to me what this means or why you'd do this? If you have full observability the flow-based methods severely underperform PPO with a gaussian parameterization.
- There's no sensitivity analysis on $\lambda$ where I can imagine the optimal value varies greatly dependent on the magnitude of the return and other properties of the environment.
- The results overall are underwhelming and there's many more principled methods from the generative modelling literature that could give better few-step policies.

Overall, the only contribution of this work is to point out that the coupling of noise in FPO can be viewed under the lens of reflow with the rest of the paper not being in a state suitable for a conference submission.

---

> ### Author Rebuttal · Authors · 2026-03-31
>
> We hope the clarifications below, together with the additional results on our anonymous website ([supplementary link](https://anonymous.4open.science/w/icml_rebuttal-1E27/)), especially Extra Table 1 and Extra Table 2, help address these concerns.
>
>
> ## Theoretical Clarification
>
> We respectfully disagree with the reviewer’s assessment that our theoretical premise is false. The critique appears to conflate two fundamentally different concepts: the computational graph used for backpropagation and the functional sensitivity of the on-policy data distribution under a parameter perturbation $\theta \mapsto \theta + \delta\theta$. In the ReFPO framework, the endpoint action $a_1$ is generated on-policy rather than being drawn from a fixed dataset as in regular CFM. This means $a_1 \sim \pi_\theta(\cdot | o_t)$, and consequently, the interpolated state $a_\tau = \alpha_\tau a_1 + \sigma_\tau \epsilon$ is inherently dependent on $\theta$ at the level of our perturbation analysis. When policy parameters change, the induced trajectory distribution shifts, which makes the term $\delta a_\tau$ generally non-zero. The use of a stop-gradient in the implementation is a choice of gradient estimator; it does not imply that the generated samples are independent of the parameters that produced them. This distinction is vital: while this dependence vanishes in offline settings with fixed data, it is the defining characteristic of on-policy RL where the policy induces its own training distribution. Therefore, our ODE sensitivity analysis captures a genuine instability mechanism arising from flow curvature.
>
> ## Question 1
>
> Regarding the empirical methodology, ReFPO was evaluated using five seeds per environment, which is the standard protocol for benchmarks in the DM Control  MuJoCo Playground [1]. In Table 1, while the 10-step rewards show some overlap in confidence intervals, it is crucial to highlight that ReFPO's 1-step rewards significantly outperform the FPO baseline with little overlap in confidence regions. Furthermore, even in the 10-step comparison where intervals overlap, ReFPO consistently yields a higher mean reward and similar confidence intervals, representing a stable and meaningful performance gain over the baseline.
>
> ## Question 2
>
> We use sparse conditioning to create a controlled partial-observability setting. In other words, we intentionally reduce the available observation information so that we can test whether the proposed flow-policy regularization remains effective when the policy must act under limited information. This is a realistic and challenging regime for humanoid control, where full observability is often unavailable in practice due to occlusion, imperfect sensing, or incomplete state measurements. At the same time, this setting helps differentiate the strengths of different policy classes: PPO is very strong in fully observed settings, whereas ReFPO is better suited to multimodal and sparse-information regimes, while also enabling strong 1-step action generation.
>
>
> ## Question 3
>
> A detailed sensitivity analysis for the regularization coefficient $\lambda$ is already provided in Table 1 for the MuJoCo Playground environments. We found that a coefficient of $\lambda=0.04$ provides the optimal balance between path straightness and policy expressivity. While the optimal value may vary slightly based on specific environment properties or reward scales, our results demonstrate that maintaining a relatively fixed ratio between the geometric regularizer and the original FPO objective is sufficient to stabilize training and enhance flow performance across different tasks.
>
> ## Question 4
>
> Our results in Table 1 and Figure 3 clearly demonstrate that ReFPO dramatically improves 1-step denoising capabilities, with 1-step performance often matching or even exceeding high-fidelity multi-step results. Our analysis suggests that for "straight" rectified flows, multi-step integration can actually accumulate discretization errors in simpler movements, whereas a single pass provides a cleaner and more coherent control signal.
> Furthermore, while existing 1-step flow methods often require auxiliary teacher models for distillation,eg. SlimFlow[2] FQL[3].Some rely on computationally expensive reparameterized gradients,eg. FPMD[4], MVP[5] and FlowRL[6].ReFPO achieves its performance by reusing the CFM loss already computed in the FPO pipeline. This enables high-performance 1-step generation via a single-line code change with zero additional computational overhead.
>
> [1]MuJoCo Playground, RSS 2025
>
> [2]SlimFlow: Training Smaller One-Step Diffusion Models with Rectified Flow, ECCV 2024
>
> [3]Flow Q-Learning, ICML 2025
>
> [4]One-Step Flow Policy Mirror Descent, Arxiv 2025
>
> [5]MEAN FLOW POLICY WITH INSTANTANEOUS VELOCITY CONSTRAINT FOR ONE-STEP ACTION GENERATION, ICLR 2026
>
> [6]FlowRL: Matching Reward Distributions for LLM Reasoning, ICLR 2026

---

> > ### Author Rebuttal · Reviewer_EFJy · 2026-04-03
> >
> > The author's rebuttal is not convincing. Specifically, you are conflating two distinct levels of analysis. FPO follows the same structure as PPO:
> >
> > 1. Collect a batch of trajectories using $\theta_{\mathrm{old}}$ storing $(o_t, a_t, \tau_i, \epsilon_i)$
> > 2. Run multiple optimization epochs on this fixed batch, updating $\theta$
> >
> > During step 2 -- which is where the clipping condition matters and where the sensitivity analysis applies -- the data $(o_t, a_t, \epsilon_i, \tau_i)$ is frozen. The interpolants $a_{\tau, t} = \alpha_\tau a_t + \sigma_\tau \epsilon$ don't change as $\theta$ changes. The question the sensitivity analysis needs to answer is: "how much does the CFM loss on this fixed batch change as I take gradient steps from $\theta_{\mathrm{old}}$ to $\theta$?". The answer here is $\mathcal{O}(|| \delta \theta ||)$, it has no dependence on $\partial_a v \delta a_{\tau}$ as claimed.
> >
> > The authors are correct that $a_1 \sim \pi_\theta(\cdot\mid o_t)$ at the distribution level, but this dependence manifests between data collection phases, not within the optimization loop on a given batch. The authors frame the stop-gradient as just a "choice of gradient estimator", implying the true functional dependence still matters. But the stop-gradient reflects a fundamental algorithmic property: FPO optimizes a surrogate objective constructed from fixed data, exactly like PPO does. We're not taking a path gradient through the sampling procedure where your sensitivity analysis would apply.
> >
> > The authors even seem to acknowledge this on L647 in the appendix stating: "Although the CFM loss in the algorithm is evaluated on explicitly constructed interpolants $a_{\tau, t}$ to analyze sensitivity we interpret the learned velocity field $v_{\theta_{\mathrm{old}}}$ as inducing a continuous flow via the ODE...". This sleight of hand is not the algorithm you propose or implement.
> >
> > Although I believe there is a simple theoretical motivation for the additional CFM loss proposed herein the current analysis and presentation of the paper is not suitable for acceptance. Ultimately, I maintain my decision to reject the paper.

---

> > > ### Author Response · Authors · 2026-04-07
> > >
> > > We thank the reviewer for this careful comment. We agree that our previous presentation was misleading: in the fixed-batch inner loop, the interpolants
> > >
> > > $$a_{\tau,t} = \alpha_\tau a_t + \sigma_\tau \epsilon_i$$
> > >
> > > are explicitly constructed and do not depend on $\theta$, so $\delta a_\tau$ is not part of the computational graph. Our earlier derivation therefore mixed the inner-loop optimization level with a flow-sensitivity argument, and we appreciate the reviewer for pointing this out. We will remove the incorrect statements and derivation from the final version.
> > >
> > > Importantly, our empirical conclusion does not rely on that incorrect derivation. On our anonymous supplementary page ([supplementary link](https://anonymous.4open.science/w/icml_rebuttal-1E27/)), **Extra Figure 1** and **Extra Table 2** show a pattern: vanilla FPO exhibits simultaneous explosions in Jacobian norm and policy ratio, while ReFPO substantially reduces both; ReFPO also improves 1-step performance. Thus, ReFPO is a low-overhead modification that improves both 1-step action quality and training stability. This correction narrows our explanation, but it does not change the algorithm, regularizer, or experiments, so it does not weaken the paper's core contribution.
> > >
> > > **1. Why ReFPO improves 1-step inference.**
> > > At rollout time, the policy action is produced by integrating the learned flow ODE
> > >
> > > $$\dot x(\tau) = v_\theta(x(\tau), \tau; o).$$
> > >
> > > For the exact trajectory, a one-step Taylor expansion gives
> > >
> > > $$x(\tau-h) = x(\tau) - h\,v_\theta(x(\tau), \tau; o) + \frac{h^2}{2}\left(\partial_\tau v_\theta + J_x v_\theta \, v_\theta\right)\big|_{(x(\tau), \tau; o)} + \mathcal O(h^3).$$
> > >
> > > Thus Euler error is controlled by $\partial_\tau v_\theta$ and $J_x v_\theta v_\theta$, while the global error constant worsens with
> > >
> > > $$L = \sup_{x,\tau}\|J_x v_\theta(x, \tau; o)\|_{\mathrm{op}}.$$
> > >
> > > When $\|J_x v_\theta\|$ is large, the flow path is highly curved and coarse discretization is inaccurate; a 1-step solver is then more error-prone than a multi-step solver. ReFPO regularizes the flow toward straighter trajectories, reducing this sensitivity and making 1-step inference much closer to multi-step rollout.
> > >
> > > **2. Why ReFPO improves training stability.**
> > > Inside one PPO-style optimization phase, the relevant object is the fixed-batch objective as a function of parameters, not a path derivative through sampled interpolants. On a frozen batch $(o_t, a_t, \tau_i, \epsilon_i)$, define
> > >
> > > $$\widehat L_{\mathrm{batch}}(\theta) = \frac{1}{N}\sum_{i=1}^N \|v_\theta(z_i, \tau_i; o_t) - (a_t - \epsilon_i)\|^2,$$
> > >
> > > with
> > >
> > > $$z_i = \alpha_{\tau_i} a_t + \sigma_{\tau_i}\epsilon_i.$$
> > >
> > > Expanding around $\theta_{old}$ gives
> > >
> > > $$\widehat L_{\mathrm{batch}}(\theta_{\mathrm{old}} + \delta\theta) = \widehat L_{\mathrm{batch}}(\theta_{\mathrm{old}}) + \langle \nabla_\theta \widehat L_{\mathrm{batch}}(\theta_{\mathrm{old}}), \delta\theta \rangle + \mathcal O(\|\delta\theta\|^2),$$
> > >
> > > with first-order term
> > >
> > > $$\nabla_\theta \widehat L_{\mathrm{batch}}(\theta_{\mathrm{old}}) = \frac{2}{N}\sum_{i=1}^N \left(\partial_\theta v_{\theta_{\mathrm{old}}}(z_i, \tau_i; o_t)\right)^T \left(v_{\theta_{\mathrm{old}}}(z_i, \tau_i; o_t) - (a_t - \epsilon_i)\right).$$
> > >
> > > This gives the proxy ratio
> > >
> > > $$\rho(\theta) = \exp\left(\widehat L_{\mathrm{batch}}(\theta_{\mathrm{old}}) - \widehat L_{\mathrm{batch}}(\theta)\right),$$
> > >
> > > for which
> > >
> > > $$\log \rho(\theta_{\mathrm{old}} + \delta\theta) = - \langle \nabla_\theta \widehat L_{\mathrm{batch}}(\theta_{\mathrm{old}}), \delta\theta \rangle + \mathcal O(\|\delta\theta\|^2).$$
> > >
> > > Moreover,
> > >
> > > $$\lVert \nabla_\theta \widehat L_{\mathrm{batch}}(\theta_{\mathrm{old}}) \rVert \le \frac{2}{N}\sum_{i=1}^N \lVert \partial_\theta v_{\theta_{\mathrm{old}}}(z_i, \tau_i; o_t) \rVert_{\mathrm{op}} \cdot \lVert v_{\theta_{\mathrm{old}}}(z_i, \tau_i; o_t) - (a_t - \epsilon_i) \rVert.$$
> > >
> > > To first order, ratio explosion is driven by large parameter sensitivity, large residual, or both. In the straight-flow limit, the target velocity along each interpolation path is closer to a nearly constant vector $(a_t - \epsilon_i)$ rather than a highly state-dependent field. This makes the target easier to fit, reducing the residual, and requires less state-dependent bending, which may also reduce parameter sensitivity. The result is a smaller fixed-batch gradient, smaller first-order variation in $\log \rho$, and therefore fewer clipping events. We present this as a mechanism consistent with the evidence, not as a theorem equating input-space and parameter-space Jacobians.
> > >
> > > In the final version, we will make this distinction explicit: the Jacobian-based geometric argument is for rollout-time 1-step accuracy, while the fixed-batch Taylor argument explains why simplifying the fitted flow can stabilize the PPO-style inner loop. Together with the supplementary evidence above, this supports ReFPO as a useful improvement for stable and efficient 1-step flow-policy learning.

---

### Decision · Program_Chairs · 2026-04-30

**Decision:**

Reject

**Comment:**

This paper proposes a modification to the RL algorithm that learns neural ODE policies known as flow matching with policy gradients -- the addition of a regulariser encouraging straight paths, which allows accurate integration of the learnt dynamics in fewer steps. While the reviewers appreciated the simplicity and effectiveness of the idea, they noted a number of weaknesses, and the following two recurring concerns did not receive responses that were found satisfactory by the reviewers:
- correctness of some theoretical claims and validity of the assumptions (EFJy, KzDd);
- experiment setup: completeness of the analysis of components of the algorithm, empirical backing for the stated motivations (all).

To the first point, I would add that Figure 1 makes the *incorrect* claim that iterative rectification converges optimal transport -- it should converge to a flow with straight noncrossing integral curves, but not necessarily to the dynamic OT. To enforce dynamic OT one needs other objectives, such as the ones in [Tong et al., TMLR, arXiv:2302.00482] and [Pooladian et al., ICML'23, arXiv:2304.14772]. (Of course, it is straightness and not OT that is needed for few-step integration.)

I tend to agree with EFJy's view that the contribution is somewhat thin, but believe the paper has promise and hope that an improved version with all the reviewers' comments addressed will be submitted to another venue.